# Imitation Learning via Off-Policy Distribution Matching

**Ilya Kostrikov,**[*] **Ofir Nachum, Jonathan Tompson**
Google Research
`{kostrikov, ofirnachum, tompson}@google.com`

## Abstract

When performing imitation learning from expert demonstrations, distribution matching is a popular approach, in which one alternates between estimating distribution ratios and then using these ratios as rewards in a standard reinforcement learning (RL) algorithm. Traditionally, estimation of the distribution ratio requires on-policy data, which has caused previous work to either be exorbitantly data-inefficient or alter the original objective in a manner that can drastically change its optimum. In this work, we show how the original distribution ratio estimation objective may be transformed in a principled manner to yield a completely off-policy objective. In addition to the data-efficiency that this provides, we are able to show that this objective also renders the use of a separate RL optimization unnecessary. Rather, an imitation policy may be learned directly from this objective without the use of explicit rewards. We call the resulting algorithm *ValueDICE* and evaluate it on a suite of popular imitation learning benchmarks, finding that it can achieve state-of-the-art sample efficiency and performance.[1]

## 1 Introduction

Reinforcement learning (RL) is typically framed as learning a behavior policy based on reward feedback from trial-and-error experience. Accordingly, many successful demonstrations of RL often rely on carefully handcrafted rewards with various bonuses and penalties designed to encourage intended behavior (Nachum et al., 2019a; Andrychowicz et al., 2018). In contrast, many real-world behaviors are easier to demonstrate rather than devise explicit rewards. This realization is at the heart of *imitation learning* (Ho & Ermon, 2016; Ng et al.; Pomerleau, 1989), in which one aims to learn a behavior policy from a set of expert demonstrations – logged experience data of a near-optimal policy interacting with the environment – without explicit knowledge of rewards.

Distribution matching via adversarial learning, or Adversarial Imitation Learning (AIL), has recently become a popular approach for imitation learning (Ho & Ermon, 2016; Fu et al., 2017; Ke et al., 2019; Kostrikov et al., 2019). These methods interpret the states and actions provided in the expert demonstrations as a finite sample from a target distribution. Imitation learning can then be framed as learning a behavior policy which minimizes a divergence between this target distribution and the state-action distribution induced by the behavior policy interacting with the environment. As derived by Ho & Ermon (2016), this divergence minimization may be achieved by iteratively performing two alternating steps, reminiscent of GAN algorithms (Goodfellow et al., 2014). First, one estimates the density ratio of states and actions between the target distribution and the behavior policy. Then, these density ratios are used as rewards for a standard RL algorithm, and the behavior policy is updated to maximize these cumulative rewards (data distribution ratios).

The main limitation of current distribution matching approaches is that estimating distribution density ratios (the first step of every iteration) typically requires samples from the behavior policy distribution. This means that every iteration – every update to the behavior policy – requires new interactions with the environment, precluding the use of these algorithms in settings where interactions with the environment are expensive and limited. Several papers attempt to relax this *on-policy*

---

[*]Also at NYU.

[1]Code to reproduce our results is available at `https://github.com/google-research/google-research/tree/master/value_dice`.

requirement and resolve the sample inefficiency problem by designing *off-policy* imitation learning algorithms, which may take advantage of past logged data, usually in the form of a replay buffer (Kostrikov et al., 2019; Sasaki et al., 2019). However, these methods do so by altering the original divergence minimization objective to measure a divergence between the target expert distribution and the replay buffer distribution. Accordingly, there is no guarantee that the learned policy will recover the desired target distribution.

In this work, we introduce an algorithm for imitation learning that, on the one hand, performs divergence minimization as in the original AIL methods, while on the other hand, is completely off-policy. We begin by providing a new formulation of the minimum divergence objective that avoids the use of any explicit on-policy expectations. While this objective may be used in the traditional way to estimate data distribution ratios that are then input to an RL algorithm, we go further to show how the specific form of the derived objective renders the use of a separate RL optimization unnecessary. Rather, gradients of the minimum divergence objective with respect to behavior policy may be computed directly. This way, an imitating behavior policy may be learned to minimize the divergence without the use of explicit rewards. We call this stream-lined imitation learning algorithm ValueDICE. In addition to being simpler than standard imitation learning methods, we show that our proposed algorithm is able to achieve state-of-the-art performance on a suite of imitation learning benchmarks.

## 2 BACKGROUND

We consider environments represented as a Markov Decision Process (MDP) (Puterman, 2014), defined by the tuple, $(\mathcal{S}, \mathcal{A}, p_0(s), p(s'|s, a), r(s, a), \gamma)$, where $\mathcal{S}$ and $\mathcal{A}$ are the state and action space, respectively, $p_0(s)$ is an initial state distribution, $p(s'|s, a)$ defines environment dynamics represented as a conditional state distribution, $r(s, a)$ is a reward function, and $\gamma$ is a return discount factor. A behavior policy $\pi(\cdot|\cdot)$ interacts with the environment to yield experience $(s_t, a_t, r_t, s_{t+1})$, for $t = 0, 1, \ldots$, where $s_0 \sim p_0(\cdot)$, $a_t \sim \pi(\cdot|s_t)$, $s_{t+1} \sim p(\cdot|s_t, a_t)$, $r_t = r(s_t, a_t)$. Without loss of generality, we consider infinite-horizon, non-terminating environments. In standard RL, one aims to learn a behavior policy $\pi(\cdot|s)$ to maximize cumulative rewards, based on experience gained from interacting with the environment.

In imitation learning (Pomerleau, 1989; Abbeel & Ng, 2004; Ho & Ermon, 2016), the environment reward is not observed. Rather, one has access to a set of expert demonstrations $\mathcal{D} := \{(s_k, a_k, s'_k)\}_{k=1}^N$ given by state-action-next-state transitions in the environment induced by an unknown expert policy $\pi_{\mathrm{exp}}$ and the goal is to learn a behavior policy $\pi$ which recovers $\pi_{\mathrm{exp}}$. During the learning process, in addition to the finite set of expert demonstrations $\mathcal{D}$, one may also optionally interact with the environment (in these interactions, no rewards are observed). This setting describes a number of real-world applications where rewards are unknown, such as Pomerleau (1989); Muller et al. (2006); Bojarski et al. (2016).

### 2.1 BEHAVIORAL CLONING (BC)

Supervised behavioral cloning (BC) is a popular approach for imitation learning. Given a set of expert demonstrations, a mapping of state observations to actions is fit using regression or density estimation. In the simplest case, one simply trains the behavior policy $\pi$ to minimize the negative log-likelihood of the observed expert actions:

$$\min_\pi J_{\mathrm{BC}}(\pi) := -\frac{1}{N} \sum_{k=1}^N \log \pi(a_k|s_k). \tag{1}$$

Unlike Inverse Reinforcement Learning (IRL) algorithms (e.g. GAIL (Ho & Ermon, 2016)), BC does not perform any additional policy interactions with the learning environment and hence does not suffer from the same issue of policy sample complexity. However, behavioral cloning suffers from distributional drift (Ross et al., 2011); i.e., there is no way for $\pi$ to learn how to recover if it deviates from the expert behavior to a state $\tilde{s}$ not seen in the expert demonstrations.

## 2.2 DISTRIBUTION MATCHING

The distribution matching approach provides a family of methods that are robust to distributional shift. Rather than considering the policy directly as a conditional distribution $\pi(\cdot|s)$ over actions, this approach considers the state-action distribution induced by a policy. In particular, under certain conditions (Puterman, 2014), there is a one-to-one correspondence between a policy and its state-action distribution $d^\pi$ defined as,

$$d^\pi(s, a) = (1 - \gamma) \cdot \sum_{t=0}^\infty \gamma^t p(s_t = s, a_t = a | s_0 \sim p_0(\cdot), s_t \sim p(\cdot|s_{t-1}, a_{t-1}), a_t \sim \pi(\cdot|s_t)). \quad (2)$$

By the same token, the unknown expert policy $\pi_{\text{exp}}$ also possesses a state-action distribution $d^{\text{exp}}$, and one may usually assume that the expert demonstrations $\mathcal{D} := \{(s_k, a_k, s'_k)\}_{k=1}^N$ are sampled as $(s_k, a_k) \sim d^{\text{exp}}, s'_k \sim p(\cdot|s_k, a_k)$.

Accordingly, the distribution matching approach proposes to learn $\pi$ to minimize the divergence between $d^\pi$ and $d^{\text{exp}}$. The KL-divergence is typically used to measure the discrepancy between $d^\pi$ and $d^{\text{exp}}$ (Ho & Ermon, 2016; Ke et al., 2019):

$$- D_{\text{KL}}\left(d^\pi || d^{\text{exp}}\right) = \mathbb{E}_{(s,a) \sim d^\pi}\left[\log \frac{d^{\text{exp}}(s, a)}{d^\pi(s, a)}\right]. \quad (3)$$

The use of the KL-divergence is convenient, as it may be expressed as an RL problem where rewards are given by log distribution ratios:

$$- D_{\text{KL}}\left(d^\pi || d^{\text{exp}}\right) = (1 - \gamma) \cdot \mathbb{E}_{\substack{s_0 \sim p_0(\cdot),\, a_t \sim \pi(\cdot|s_t) \\ s_{t+1} \sim p(\cdot|s_t, a_t)}}\left[\sum_{t=0}^\infty \gamma^t \log \frac{d^{\text{exp}}(s_t, a_t)}{d^\pi(s_t, a_t)}\right]. \quad (4)$$

In other words, if one has access to estimates of the distribution ratios of the two policies, then the minimum divergence problem reduces to a max-return RL problem with rewards $\tilde{r}(s, a) = \log \frac{d^{\text{exp}}(s,a)}{d^\pi(s,a)}$. Any on-policy or off-policy RL algorithm can be used to maximize the corresponding expected returns in Equation 4.

Capitalizing on this observation, Ho & Ermon (2016) and Ke et al. (2019) propose algorithms (e.g., GAIL) in which the distribution ratio is estimated using a GAN-like objective:

$$\max_{h:\mathcal{S} \times \mathcal{A} \to (0,1)} J_{\text{GAIL}}(h) := \mathbb{E}_{(s,a) \sim d^{\text{exp}}}[\log h(s, a)] + \mathbb{E}_{(s,a) \sim d^\pi}[\log(1 - h(s, a))]. \quad (5)$$

In this objective, the function $h$ acts as a *discriminator*, discriminating between samples $(s, a)$ from $d^{\text{exp}}$ and $d^\pi$. The optimal discriminator satisfies,

$$\log h^*(s, a) - \log(1 - h^*(s, a)) = \log \frac{d^{\text{exp}}(s, a)}{d^\pi(s, a)}, \quad (6)$$

and so the distribution matching rewards may be computed as $\tilde{r}(s, a) = \log h^*(s, a) - \log(1 - h^*(s, a))$. In practice, the discriminator is not fully optimized, and instead gradient updates to the discriminator and policy are alternated.

These prior distribution matching approaches possess two limitations which we will resolve with our proposed ValueDICE algorithm:

- **On-policy.** Arguably the main limitation of these prior approaches is that they require access to on-policy samples from $d^\pi$. While off-policy RL can be used for learning $\pi$, optimizing the discriminator $h$ necessitates having on-policy samples (the second expectation in Equation 5). Thus, in practice, GAIL requires a prohibitively large number of environment interactions, making it unfeasible for use in many real-world applications. Attempts to remedy this, such as Discriminator-Actor-Critic (DAC) (Kostrikov et al., 2019), often do so via ad-hoc methods; for example, changing the on-policy expectation $\mathbb{E}_{(s,a) \sim d^\pi}[\log(1 - h(s, a))]$ in Equation 5 to an expectation over the replay buffer $\mathbb{E}_{(s,a) \sim d^{\text{RB}}}[\log(1 - h(s, a))]$. While DAC achieves good empirical results, it does not guarantee distribution matching of $\pi$ to $\pi_{\text{exp}}$, especially when $d^{\text{RB}}$ is far from $d^\pi$.
- **Separate RL optimization.** Prior approaches require iteratively taking alternating steps: first estimate the data distribution ratios using the GAN-like objective, then input these into an RL optimization and repeat. The use of a separate RL algorithm introduces complexity to any implementation of these approaches, with many additional design choices that need to be made and more function approximators to learn (e.g., value functions). Our introduced ValueDICE will be shown to not need a separate RL optimization.

## 3    OFF-POLICY FORMULATION OF THE KL-DIVERGENCE

As is standard in distribution matching, we begin with the KL-divergence between the policy state-action occupancies and the expert. However, in contrast to the form used in Equation 4 or 5, we use the Donsker-Varadhan representation (Donsker & Varadhan, 1983) given by,

$$-D_{\mathrm{KL}}\left(d^{\pi}||d^{\exp}\right) = \min_{x:\mathcal{S}\times\mathcal{A}\to\mathbb{R}} \log \mathbb{E}_{(s,a)\sim d^{\exp}}[e^{x(s,a)}] - \mathbb{E}_{(s,a)\sim d^{\pi}}[x(s,a)]. \tag{7}$$

Similar to Equation 5, this dual representation of the KL has a property that is important for imitation learning. The optimal $x^*$ is equal to the log distribution ratio (plus a constant):[2]

$$x^*(s,a) = \log \frac{d^{\pi}(s,a)}{d^{\exp}(s,a)} + C. \tag{8}$$

In our considered infinite-horizon setting, the constant does not affect optimality and so we will ignore it (take $C = 0$). If one were to take a GAIL-like approach, they could use this form of the KL to estimate distribution matching rewards given by $\tilde{r}(s,a) = -x^*(s,a)$, and these could then be maximized by any standard RL algorithm. However, there is no clear advantage of this objective over GAIL since it still relies on an expectation with respect to on-policy samples from $d^{\pi}$.

To make this objective practical for off-policy learning, we take inspiration from derivations used in DualDICE (Nachum et al., 2019b), and perform the following change of variables:[3]

$$x(s,a) = \nu(s,a) - \mathcal{B}^{\pi}\nu(s,a), \tag{9}$$

where $\mathcal{B}^{\pi}$ is the expected Bellman operator with respect to policy $\pi$ and zero reward:

$$\mathcal{B}^{\pi}\nu(s,a) = \gamma\mathbb{E}_{s'\sim p(\cdot|s,a),a'\sim\pi(\cdot|s')}[\nu(s',a')]. \tag{10}$$

This change of variables is explicitly chosen to take advantage of the linearity of the second expectation in Equation 7. Specifically, the representation becomes,

$$-D_{\mathrm{KL}}\left(d^{\pi}||d^{\exp}\right) = \min_{\nu:\mathcal{S}\times\mathcal{A}\to\mathbb{R}} \log \mathbb{E}_{(s,a)\sim d^{\exp}}[e^{\nu(s,a)-\mathcal{B}^{\pi}\nu(s,a)}] - \mathbb{E}_{(s,a)\sim d^{\pi}}[\nu(s,a)-\mathcal{B}^{\pi}\nu(s,a)], \tag{11}$$

where the second expectation conveniently telescopes and reduces to an expectation over initial states (see Nachum et al. (2019b) for details):

$$\min_{\nu:\mathcal{S}\times\mathcal{A}\to\mathbb{R}} J_{\mathrm{DICE}}(\nu) := \log \mathbb{E}_{(s,a)\sim d^{\exp}}[e^{\nu(s,a)-\mathcal{B}^{\pi}\nu(s,a)}] - (1-\gamma)\cdot \mathbb{E}_{\substack{s_0\sim p_0(\cdot),\\a_0\sim\pi(\cdot|s_0)}}[\nu(s_0,a_0)]. \tag{12}$$

Thus we achieve our *ValueDICE*[4] objective. It allows us to express the KL-divergence between $d^{\pi}$ and $d^{\exp}$ in terms of an objective over a 'value-function' $\nu$ expressed in an off-policy manner, with expectations over expert demonstrations $d^{\exp}$ and initial state distribution $p_0(\cdot)$.

It is clear that the derived objective in Equation 12 possesses two key characteristics missing from prior distribution matching algorithms: First, the objective does not rely on access to samples from the on-policy distribution $d^{\pi}$, and so may be used in more realistic, off-policy settings. Second, the objective describes a proper divergence between $d^{\pi}$ and $d^{\exp}$, as opposed to estimating a divergence between $d^{\mathrm{RB}}$ and $d^{\exp}$, and thus avoids poor behavior when $d^{\mathrm{RB}}$ is far from $d^{\pi}$. In the following section, we will go further to show how the objective in Equation 12 also renders the use of a separate RL optimization unnecessary.

## 4    VALUEDICE: IMITATION LEARNING WITH IMPLICIT REWARDS

Although it is standard in distribution matching to have separate optimizations for estimating the distribution ratios and learning a policy, in our case this can be mitigated. Indeed, looking at our

---

[2]This result is easy to derive by setting the gradient of the Donsker-Varadhan representation to zero and solving for $x^*$.

[3]This change of variables is valid when one assumes $\log d^{\pi}(s,a)/d^{\exp}(s,a) \in \mathcal{K}$ for all $s \in \mathcal{S}, a \in \mathcal{A}$, where $\mathcal{K}$ is some bounded subset of $\mathbb{R}$, and $x$ is restricted to the family of functions $\mathcal{S}\times\mathcal{A}\to\mathcal{K}$.

[4]DICE (Nachum et al., 2019b) is an abbreviation for discounted distribution correction estimation.

formulation of the KL in Equation 12, we see that gradients of this objective with respect to $\pi$ may be easily computed. Specifically, we may express the distribution matching objective for $\pi$ as a max-min optimization:

$$\max_{\pi} \min_{\nu:\mathcal{S}\times\mathcal{A}\to\mathbb{R}} J_{\text{DICE}}(\pi,\nu) := \log \mathbb{E}_{(s,a)\sim d^{\text{exp}}}[e^{\nu(s,a)-\mathcal{B}^\pi \nu(s,a)}] - (1-\gamma) \cdot \mathbb{E}_{\substack{s_0 \sim p_0(\cdot), \\ a_0 \sim \pi(\cdot|s_0)}}[\nu(s_0,a_0)]. \quad (13)$$

If the inner objective over $\nu$ is sufficiently optimized, the gradients of $\pi$ may be computed directly (Bertsekas, 1999), noting that,

$$\frac{\partial}{\partial \pi} e^{\nu(s,a)-\mathcal{B}^\pi \nu(s,a)} = -\gamma \cdot e^{\nu(s,a)-\mathcal{B}^\pi \nu(s,a)} \cdot \mathbb{E}_{s'\sim T(s,a),a'\sim\pi(s')}[\nu(s',a')\nabla \log \pi(a'|s')], \quad (14)$$

$$\frac{\partial}{\partial \pi} \mathbb{E}_{s_0\sim p_0(\cdot),a_0\sim\pi(\cdot|s_0)}[\nu(s_0,a_0)] = \mathbb{E}_{s_0\sim p_0(\cdot),a_0\sim\pi(\cdot|s_0)}[\nu(s_0,a_0)\nabla \log \pi(a_0|s_0)]. \quad (15)$$

In continuous control environments when $\pi$ is parameterized by a Gaussian and $\nu$ is a neural network, one may use the re-parameterization trick (Haarnoja et al., 2018) to compute gradients of the $\nu$-values with respect to policy mean and variance directly as opposed to computing $\nabla \log \pi(a|s)$. Please see the appendix for a full pseudocode implementation of ValueDICE. We note that in practice, as in GAIL, we do not train $\nu$ until optimality but rather alternate $\nu$ and $\pi$ updates.

The mechanics of learning $\pi$ according to the ValueDICE objective are straightforward, but what is the underlying reason for this more streamlined policy learning? How does it relate the standard protocol of alternating data distribution estimation with RL optimization? To better understand this, we consider the form of $\nu$ when it is completely optimized. If we consider the original change of variables (Equation 9) and optimality characterization (Equation 8) we have,

$$\nu^*(s,a) - \mathcal{B}^\pi \nu^*(s,a) = x^*(s,a) = \log \frac{d^\pi(s,a)}{d^{\text{exp}}(s,a)}. \quad (16)$$

From this characterization of $\nu^*$, we realize that $\nu^*$ is a sort of Q-value function: $\nu^*(s,a)$ is the future discounted sum of rewards $\tilde{r}(s,a) := \log \frac{d^\pi(s,a)}{d^{\text{exp}}(s,a)}$ when acting according to $\pi$. The gradients for $\pi$ then encourage the policy to choose actions which *minimize* $\nu^*(s,a)$, i.e., *maximize* future discounted log ratios $\log \frac{d^{\text{exp}}(s,a)}{d^\pi(s,a)}$. Thus we realize that the objective for $\pi$ in ValueDICE performs exactly the RL optimization suggested by Equation 4. The streamlined nature of ValueDICE comes from the fact that the value function $\nu$ (which would traditionally need to be learned as a critic in a separate actor-critic RL algorithm) is learned directly from the same objective as that used for distribution matching.

Thus, in addition to estimating a proper divergence between $d^\pi$ and $d^{\text{exp}}$ in an off-policy manner, ValueDICE also greatly simplifies the implementation of distribution matching algorithms. There is no longer a need to use a separate RL algorithm for learning $\pi$, and moreover, the use of $\nu$ as a value function removes any use of explicit rewards. Instead, the objective and implementation are only in terms of policy $\pi$ and function $\nu$.

## 5 SOME PRACTICAL CONSIDERATIONS

In order to make use of the ValueDICE objective (Equation 13) in practical scenarios, where one does not have access to $d^{\text{exp}}$ or $p_0(\cdot)$ but rather only limited finite samples, we perform several modifications.

### 5.1 EMPIRICAL EXPECTATIONS

The objective in Equation 13 contains three expectations:

1. An expectation over $d^{\text{exp}}$ (the first term of the objective). Note that this expectation has a logarithm outside of it, which would make any mini-batch approximations of the gradient of this expectation biased.

2. An expectation over $p_0(\cdot)$ (the second term of the objective). This term is linear, and so is very amenable to mini-batch optimization.

3. An expectation over the environment transition $p(\cdot|s,a)$ used to compute $\mathcal{B}^\pi\nu(s,a)$. This expectation has a log-expected-exponent applied to it, so its mini-batch approximated gradient would be biased in general.

For the first expectation, previous works have suggested a number of remedies to reduce the bias of mini-batch gradients, such as maintaining moving averages of various quantities (Belghazi et al., 2018). In the setting we considered, we found this to have a negligible effect on performance. In fact, simply using the biased mini-batched gradients was sufficient for good performance, and so we used this for our experiments.

For the second expectation, we use standard mini-batch gradients, which are unbiased. Although initial state distributions are usually not used in imitation learning, it is easy to record initial states as they are observed, and thus have access to an empirical sample from $p_0$. Furthermore, as detailed in Section 5.3, it is possible to modify the initial state distribution used in the objective without adverse effects.

Finally, for the third expectation, previous works have suggested the use of Fenchel conjugates to remove the bias (Nachum et al., 2019b). In our case, we found this unnecessary and instead use a biased estimate based on the single sample $s' \sim p(\cdot|s,a)$. This naive approach was enough to achieve good performance on the benchmark domains we considered.

In summary, the empirical form of the objective is given by,

$$
\hat{J}_{\mathrm{DICE}}(\pi,\nu) =
$$

$$
\mathbb{E}_{\substack{\mathrm{batch}(\mathcal{D})\sim\mathcal{D},\\ \mathrm{batch}(p_0)\sim\hat{p}_0}} \left[ \log \mathbb{E}_{\substack{s,a,s'\sim\mathrm{batch}(\mathcal{D}),\\ a'\sim\pi(\cdot|s')}} \left[ e^{\nu(s,a)-\gamma\nu(s',a')} \right] - (1-\gamma)\cdot \mathbb{E}_{\substack{s_0\sim\mathrm{batch}(p_0),\\ a_0\sim\pi(\cdot|s_0)}} \left[ \nu(s_0,a_0) \right] \right], \quad (17)
$$

where $\mathrm{batch}(\mathcal{D})$ is a mini-batch from $\mathcal{D}$ and $\mathrm{batch}(p_0)$ is a mini-batch from the recorded initial states $\hat{p}_0$.

## 5.2 REPLAY BUFFER REGULARIZATION

The original ValueDICE objective uses only expert samples and the initial state distribution. In practice, the number of expert samples may be small and lack diversity, hampering learning. In order to increase the diversity of samples used for training, we consider an alternative objective, with a controllable regularization based on experience in the replay buffer:

$$
J_{\mathrm{DICE}}^{\mathrm{mix}}(\pi,\nu) := \log \mathbb{E}_{(s,a)\sim d^{\mathrm{mix}}}[e^{\nu(s,a)-\mathcal{B}^\pi\nu(s,a)}] - (1-\alpha)(1-\gamma)\cdot \mathbb{E}_{\substack{s_0\sim p_0(\cdot),\\ a_0\sim\pi(\cdot|s_0)}} \left[ \nu(s_0,a_0) \right]
$$

$$
- \alpha \mathbb{E}_{(s,a)\sim d^{\mathrm{RB}}} \left[ \nu(s,a) - \mathcal{B}^\pi\nu(s,a) \right], \quad (18)
$$

where $d^{\mathrm{mix}}(s,a) = (1-\alpha)d^{\mathrm{exp}}(s,a) + \alpha d^{\mathrm{RB}}(s,a)$.

The main advantage of this formulation is that it introduces $\nu$-values into the objective on samples that are outside the given expert demonstrations. Thus, if $\pi$ deviates from the expert trajectory, we will still be able to learn optimal actions that return the policy back to the expert behavior. At the same time, one can verify that in this formulation the optimal $\pi$ still matches $\pi_{\mathrm{exp}}$, unlike other proposals for incorporating a replay buffer distribution (Kostrikov et al., 2019). Indeed, the objective in Equation 18 corresponds to the Donsker-Varadhan representation,

$$
- D_{\mathrm{KL}}((1-\alpha)d^\pi + \alpha d^{\mathrm{RB}} \,||\, (1-\alpha)d^{\mathrm{exp}} + \alpha d^{\mathrm{RB}}) =
$$

$$
\min_{x:\mathcal{S}\times\mathcal{A}\to\mathbb{R}} \log \mathbb{E}_{(s,a)\sim d^{\mathrm{mix}}}[e^{x(s,a)}] - (1-\alpha)\cdot\mathbb{E}_{(s,a)\sim d^\pi}\left[x(s,a)\right] - \alpha\cdot\mathbb{E}_{(s,a)\sim d^{\mathrm{RB}}}\left[x(s,a)\right], \quad (19)
$$

and so the optimal values of $\nu^*$ satisfy,

$$
\nu^*(s,a) - \mathcal{B}^\pi\nu^*(s,a) = x^*(s,a) = \log \frac{(1-\alpha)d^\pi(s,a) + \alpha d^{\mathrm{RB}}(s,a)}{(1-\alpha)d^{\mathrm{exp}}(s,a) + \alpha d^{\mathrm{RB}}(s,a)}. \quad (20)
$$

Therefore, the global optimality of $\pi = \pi_{\mathrm{exp}}$ is unaffected by any choice of $\alpha < 1$. We note that in practice we use a small value $\alpha = 0.1$ for regularization.

## 5.3 INITIAL STATE SAMPLING

Recall that $d^{\exp}, d^\pi$ traditionally refer to *discounted* state-action distributions. That is, sampling from them is equivalent to first sampling a trajectory $(s_0, a_0, s_1, a_1, \ldots, s_T)$ and then sampling a time index $t$ from a geometric distribution $\text{Geom}(1 - \gamma)$ (appropriately handling samples that are beyond $T$). This means that samples far into the trajectory do not contribute much to the objective. To remedy this, we propose treating every state in a trajectory as an 'initial state.' That is, we consider a single environment trajectory $(s_0, a_0, s_1, a_1, \ldots, s_T)$ as $T$ distinct *virtual* trajectories $\{(s_t, a_t, s_{t+1}, a_{t+1}, \ldots, s_T)\}_{t=0}^{T-1}$. We apply this to both $d^{\exp}$ and $d^\pi$, so that not only does it increase the diversity of samples from $d^{\exp}$, but it also expands the initial state distribution $p_0(\cdot)$ to encompass every state in a trajectory. We note that this does not affect the optimality of the objective with respect to $\pi$, since in Markovian environments an expert policy should be expert regardless of the state at which it starts (Puterman, 2014).

## 6 RELATED WORK

In recent years, the development of Adversarial Imitation Learning has been mostly focused on on-policy algorithms. After Ho & Ermon (2016) proposed GAIL to perform imitation learning via adversarial training, a number of extensions has been introduced. Many of these applications of the AIL framework (Li et al., 2017; Hausman et al., 2017; Fu et al., 2017) maintain the same form of distribution ratio estimation as GAIL which necessitates on-policy samples. In contrast, our work presents an off-policy formulation of the same objective.

Although several works have attempted to apply the AIL framework to off-policy settings, these previous approaches are markedly different from our own. For example, Kostrikov et al. (2019) proposed to train the discriminator in the GAN-like AIL objective using samples from a replay buffer instead of samples from a policy. This changes the distribution ratio estimation to measure a divergence between the expert and the replay. Although we introduce a controllable parameter $\alpha$ for incorporating samples from the replay buffer into the data distribution objective, we note that in practice we use a very small $\alpha = 0.1$. Furthermore, by using samples from the replay buffer in *both* terms of the objective as opposed to just one, the global optimality of the expert policy is not affected.

The off-policy formulation of the KL-divergence we derive is motivated by similar techniques in DualDICE (Nachum et al., 2019b). Still, our use of these techniques provides several novelties. First, Nachum et al. (2019b) only use the divergence formulation for data distribution estimation (which is used for off-policy evaluation), assuming a fixed policy. We use the formulation for learning a policy to minimize the divergence directly. Moreover, previous works have only applied these derivations to the $f$-divergence form of the KL-divergence, while we are the first to utilize the Donsker-Varadhan form. Anecdotally in our initial experiments, we found that using the $f$-divergence form leads to poor performance. We note that our proposed objective follows a form similar to REPS (Peters et al., 2010), which also utilizes a log-average-exp term. However, policy and value learning in REPS are performed via a bi-level optimization (i.e., the policy is learned with respect to a different objective), which is distinct from our algorithm, which trains values and policy with respect to the same objective. Our proposed ValueDICE is also significant for being able to incorporate arbitrary (non-expert) data into its learning.

## 7 EXPERIMENTS

We evaluate ValueDICE in a variety of settings, starting with a simple synthetic task before continuing to an evaluation on a suite of MuJoCo benchmarks.

### 7.1 RING MDP

We begin by analyzing the behavior of ValueDICE on a simple synthetic MDP (Figure 1). The states of the MDP are organized in a ring. At each state, two actions are possible: move clockwise or counter-clockwise. We first look at the performance of ValueDICE in a situation where the expert data is sparse and does not cover all states and actions. Specifically, we provide expert demonstra-

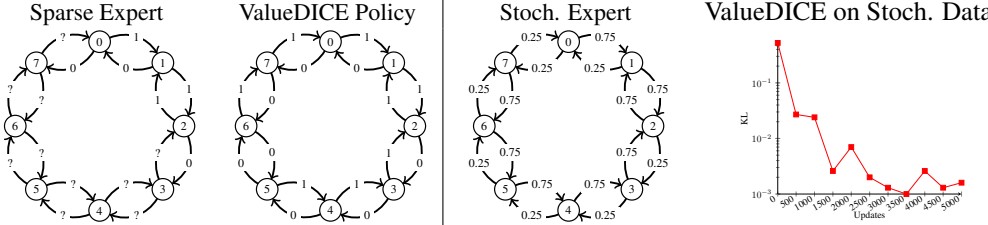

Figure 1: Results of ValueDICE on a simple Ring MDP. **Left**: The expert data is sparse and only covers states 0, 1, and 2. Nevertheless, ValueDICE is able to learn a policy on *all* states to best match the observed expert state-action occupancies (the policy learns to always go to states 1 and 2). **Right**: The expert is stochastic. ValueDICE is able to learn a policy which successfully minimizes the true KL computed between $d^\pi$ and $d^{\exp}$.

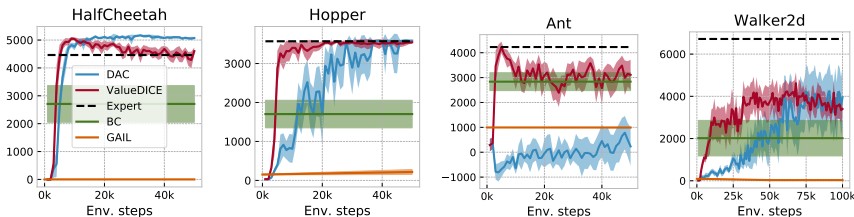

Figure 2: Comparison of algorithms given 1 expert trajectory. We use the original implementation of GAIL (Ho & Ermon, 2016) to produce GAIL and BC results.

tions which cover only states 0, 1, and 2 (see Figure 1 left). While the problem of recovering the true (unknown) expert is ill-defined, it is still possible to find a policy which recovers close to the same occupancies. Indeed, this is the policy found by ValueDICE, which chooses the appropriate actions at each state to optimally reach states 1 and 2 (and alternating between states 1 and 2 when at these states). In many practical scenarios, this is the desired outcome – if the imitating policy somehow encounters a situation which deviates from the expert demonstrations, we would like it to return to the expert behavior as fast as possible. Notably, a technique like behavioral cloning would fail to learn this optimal policy, since its learning is only based on observed expert data.

We also analyzed the behavior of ValueDICE with a stochastic expert (Figure 1 right). By using a synthetic MDP, we are able to measure the divergence $D_{\mathrm{KL}}(d^\pi || d^{\exp})$ during training. As expected, we find that this divergence decreases during ValueDICE training.

## 7.2 MuJoCo Benchmarks

We compare ValueDICE against Discriminator-Actor-Critic (DAC) (Kostrikov et al., 2019), which is the state-of-the-art in sample-efficient adversarial imitation learning, as well as GAIL (Ho & Ermon, 2016). We evaluate the algorithms on the standard MuJoCo environments using expert demonstrations from Ho & Ermon (2016). We plot the average returns for the learned policies

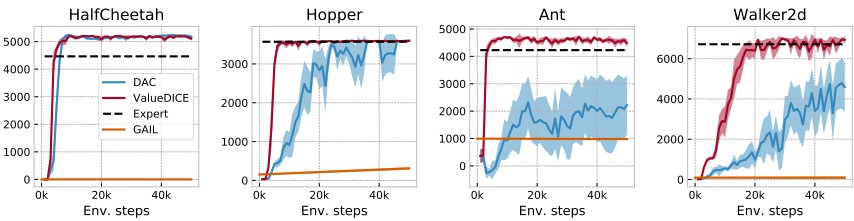

Figure 3: Comparison of algorithms given 10 expert trajectories. ValueDICE outperforms other methods. However, given this amount of data, BC can recover the expert policy as well.

(using a mean action for sampling) every 1000 environment steps using 10 episodes. We perform this procedure for 10 different seeds and compute means and standard deviations (see Fig. 2 and 3, we visualize a half of standard deviation on these plots).

We present the extremely low-data regime first. In Figure 2 we present the results of the imitation learning algorithms given only a single expert trajectory. We find that ValueDICE performs similar or better than DAC an all tasks, with the exception of Walker2d where it converges to a slightly worse policy. Notably, in this low-data regime, behavioral cloning (BC) usually cannot recover the expert policy. We also present the results of these algorithms on a larger number of expert demonstrations (Figure 3). We continue to observe strong performance of ValueDICE as well as faster convergence on all tasks. It is worth mentioning that in this large-data regime, Behavior Cloning can recover the expert performance as well. In all of these scenarios, GAIL is too sample-inefficient to make any progress.

## 8 Conclusion

We introduced ValueDICE, an algorithm for imitation learning that outperforms the state-of-the-art on standard MuJoCo tasks. In contrast to other algorithms for off-policy imitation learning, the algorithm introduced in this paper performs robust divergence minimization in a principled off-policy manner and a strong theoretical framework. To the best of our knowledge, this is also the first algorithm for adversarial imitation learning that omits learning or defining rewards explicitly and directly learns a Q-function in the distribution ratio objective directly. We demonstrate the robustness of ValueDICE in a challenging synthetic tabular MDP environment, as well as on standard MuJoCo continuous control benchmark environments, and we show increased performance over baselines in both the low and high data regimes.

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

## A  IMPLEMENTATION DETAILS

All algorithms use networks with an MLP architecture with 2 hidden layers and 256 hidden units. For discriminators, critic, $\nu$ we use Adam optimizer with learning rate $10^{-3}$ while for the actors we use the learning rate of $10^{-5}$. For the discriminator and $\nu$ networks we use gradient penalties from Gulrajani et al. (2017). We also regularize the actor network with the orthogonal regularization (Brock et al., 2018) with a coefficient $10^{-4}$. Also we perform 4 updates per 1 environment step. We handle absorbing states of the environments similarly to Kostrikov et al. (2019).

## B  ALGORITHMS

In this section, we present pseudocode for the imitation learning algorithms based on DualDICE.

---

**Algorithm 1** ValueDICE

---

**Input**: expert replay buffer $\mathcal{R}_E$

  Initialize replay buffer $\mathcal{R} \leftarrow \emptyset$

  **for** $n = 1, \dots,$ **do**

    Sample $(s, a, s')$ with $\pi_\theta$

    Add $(s, a, s')$ to the replay buffer $\mathcal{R}$

    $\{(s^{(i)}, a^{(i)}, s'^{(i)})\}_{i=1}^B \sim \mathcal{R}$              ▷ Geometric sampling

    $\{(s_0^{(i)}, s_E^{(i)}, a_E^{(i)}, s_E'^{(i)})\}_{i=1}^B \sim \mathcal{R}_E$    ▷ Geometric sampling, $s_0^{(i)}$ is a starting episode state for $s_E^{(i)}$

    $a_0^{(i)} \sim \pi_\theta(\cdot|s_0^{(i)})$, for $i = 1, \dots, B$

    $a'^{(i)} \sim \pi_\theta(\cdot|s'^{(i)})$, for $i = 1, \dots, B$

    $a_E'^{(i)} \sim \pi_\theta(\cdot|s_E'^{(i)})$, for $i = 1, \dots, B$

    Compute loss on expert data:

    $\hat{J}_{log} = \log(\frac{1}{B} \sum_{i=1}^B ((1-\alpha)e^{\nu_\psi(s_E^{(i)}, a_E^{(i)}) - \gamma\nu_\psi(s_E'^{(i)}, a_E'^{(i)})} + \alpha e^{\nu_\psi(s^{(i)}, a^{(i)}) - \gamma\nu_\psi(s'^{(i)}, a'^{(i)})}))$

    Compute loss on the replay buffer:

    $\hat{J}_{linear} = \frac{1}{B} \sum_{i=1}^B ((1-\alpha)(1-\gamma)\nu_\psi(s_0^{(i)}, a_0^{(i)}) + \alpha(\nu_\psi(s^{(i)}, a^{(i)}) - \gamma\nu_\psi(s'^{(i)}, a'^{(i)})))$

    Update $\psi \leftarrow \psi - \eta_\nu \nabla_\psi(\hat{J}_{log} - \hat{J}_{linear})$

    Update $\theta \leftarrow \psi + \eta_\pi \nabla_\theta(\hat{J}_{log} - \hat{J}_{linear})$

  **end for**

---

## C  ADDITIONAL EXPERIMENTS

We also compared ValueDICE with behavioral cloning in the offline regime, when we sample no additional transitions from the learning environment (see Figure 4). Even given only offline data, ValueDICE outperforms behavioral cloning. For behavioral cloning we used the same regularization as for actor training in ValueDICE.

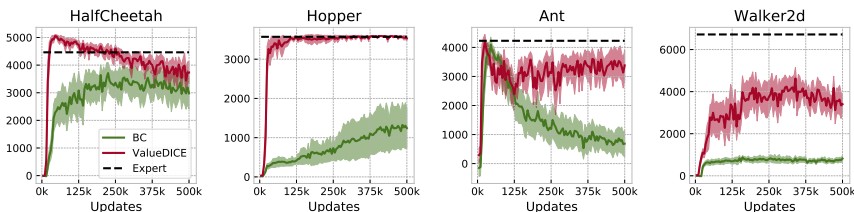

Figure 4: ValueDICE outperforms behavioral cloning given 1 trajectory even without replay regularization.

