# OpenReview forum: "Imitation Learning via Off-Policy Distribution Matching"
_ICLR.cc/2020/Conference — Accept (Poster)_

### Official Review · AnonReviewer1 · 2019-10-22
**Official Blind Review #1**

**Rating:** 6

**Review:**

The primary contribution of this paper is a principled algorithm for off-policy imitation learning. Generative Adversarial Imitation Learning (GAIL), proposed by Ho and Ermon in 2016, is an on-policy imitation learning method that (provably) minimizes the divergence between a target state-action distribution and the policy state-action distribution. Followups to this work (Kostrikov et al., 2019) show that the same algorithm can be applied to an off-policy setting (replacing the on-policy samples with samples from the replay buffer) and the method still works, but is no longer theoretically justified. I believe using importance ratios would make this approach justified as well, but Kostrikov et al. found that using importance ratios actually degrades the performance of their method (due the difficulty associated with estimating importance ratios). This paper attempts to bridge this gap: a method that is theoretically justified, and still works.

The paper takes most of its inspiration from the recently proposed DualDICE paper (Nachum et al, 2019), where the authors introduce a method for estimating discounted stationary distribution ratios (i.e. d^{\pi}/d^{D}, where \pi is some (known) policy, and D is a given dataset of experience (for example, a replay buffer)). The authors essentially apply the method proposed in DualDICE to estimate d^{\pi}/d^{exp} instead, where d^{exp} is a dataset of expert trajectories. While it would be possible to simply use this term as a reward and then run reinforcement learning, the authors note that the specific form of estimating d^{\pi}/d^{exp} allows them to instead directly a train a value function, which can then be used for updating a policy.

The authors also argue that their method reduces complexity since it does not require a separate RL optimization routine. However, I think having a separate RL optimization routine has its own advantages - it is relatively easy to implement GAIL on top of any existing RL algorithm, which makes it easy to take advantage of recent advances in RL. For the method proposed in this paper, it would not be straightforward to do so.

The authors also note that they need a number of practical modifications to their original ValueDICE objective in order to make things work (Section 5: Some practical considerations). Notably, the the original ValueDICE objective only needs access to expert samples and the initial state distribution, and does not need access to the the replay buffer samples (apart from the initial state ones). This would likely not work well in practice - similar to how behavior cloning often does worse than GAIL when learning from a small number of expert examples. In order to combat this, the authors incorporate replay buffer regularization

The authors provide experiments on  a simple synthetic "ring MDP", and on four continuous control tasks from OpenAI gym - HalfCheetah, Hopper, Ant and Walker2d. When compared to prior approaches (i.e. Kostrikov et al 2019) in the low-data regime (where behavior cloning fails), the proposed method does significantly better on one task (Ant), slightly worse on one task (walker), and about the same on two tasks (Hopper and HalfCheetah). I do notice the proposed method as being somewhat unstable though - the reward appears to be going down after reaching the max on two of the tasks - HalfCheetah and Ant. Overall, I don't thing experiments are thorough enough to demonstrate that the method is empirically better than competing approaches, but I believe that is not the main point of the paper.

Overall, my recommendation is a weak accept. It is interesting that the authors were able to get a principled method for off-policy imitation learning working (large following from prior work in Nachum et al 2019), but I don't think the method currently offers any significant practical advantages over competing methods.


**Experience Assessment:**

I have published one or two papers in this area.

**Review Assessment: Checking Correctness Of Derivations And Theory:**

I assessed the sensibility of the derivations and theory.

**Review Assessment: Checking Correctness Of Experiments:**

I carefully checked the experiments.

**Review Assessment: Thoroughness In Paper Reading:**

I read the paper thoroughly.

---

> ### Author Response · Authors · 2019-11-11
> **Re: Official Blind Review #1**
>
> Thanks for the useful feedback! Our responses are below.
>
> Regarding lack of practical advantages, since the submission we have made minor updates to our implementation. The updated experiment results (see the updated PDF) now show that ValueDICE performs better on Hopper and HalfCheetah (and generally exhibits better and more stable performance than the baselines). Our implementation updates were pretty minor and not algorithmic: we noticed that our implementation of gradient penalty was affected by the instability of tf.norm (there is tf issue for this that has been open for years https://github.com/tensorflow/tensorflow/issues/12071), and we switched our implementation to a numerically stable version of tf.norm suggested in the discussion of the issue.
>
> “This would likely not work well in practice - similar to how behavior cloning often does worse than GAIL when learning from a small number of expert examples. In order to combat this, the authors incorporate replay buffer regularization”
>
> We have updated the paper to include results of a completely offline version of ValueDICE (see Appendix C).  We show that even without any replay buffer regularization, ValueDICE can outperform BC. We believe this is the first demonstration of an imitation learning algorithm beating BC in the completely offline setting.

---

### Official Review · AnonReviewer2 · 2019-10-23
**Official Blind Review #2**

**Rating:** 6

**Review:**

This paper presents an algorithm for adversarial imitation that uses off-policy data in a principled manner, unlike prior work. The core idea is to express the KL-divergence between the policy's state-action marginal and the expert's state action marginal using the Donsker-Varadhan representation and then applying the change of variable similar to DualDICE to avoid computing the marginal of the current policy, thus getting rid of the on-policy sampling requirement. The paper then shows how the auxiliary variable (critic) added to the optimization is a value function that maximizes the corresponding induced reward in AIL methods, thus unifying the objectives for policy optimization and reward learning. The authors then present practical considerations needed in getting this formulation to work, including sampling from a replay buffer, biased sampling for the exponentiated term and avoid the double-sampling issue. Finally, the paper presents some results, which show that valueDICE is comparable to most of the other imitation learning methods.

I lean towards accepting this paper. The overall idea seems neat, however, Section 5, and the addition of a replay buffer distribution in the KL-divergence objective, though motivated enough seem to be somewhat not so principled. The experimental section is a bit weak, and I encourage the authors to strengthen this section. Also, in the case of HalfCheetah and Ant (Figure 2), valueDICE usually exhibits overfitting-like trends (the performance drops with more training), why does this happen? Can this be corrected? Overall the idea is neat, I would still say that the components very prominently exist in the literature (f-GANs, DualDICE, etc).

I would encourage the authors to add some more details experimentally and make the implementation available. For example, optimization of Bellman backup functions without target networks or delays can be unstable, solving saddle point problems can be unstable, etc. How should these factors be tuned? And overall, is the optimization of the ValueDICE objective easy?. DualDICE is known to be unstable (the DualDICE Github implementation mentions this), do similar problems arise with ValueDICE?

**Experience Assessment:**

I have read many papers in this area.

**Review Assessment: Checking Correctness Of Derivations And Theory:**

I assessed the sensibility of the derivations and theory.

**Review Assessment: Checking Correctness Of Experiments:**

I assessed the sensibility of the experiments.

**Review Assessment: Thoroughness In Paper Reading:**

I read the paper at least twice and used my best judgement in assessing the paper.

---

> ### Author Response · Authors · 2019-11-11
> **Re: Official Blind Review #2**
>
> We thank the reviewer for the close reading of the paper and the helpful feedback!
>
> We respectfully disagree that the addition of replay buffer regularization is not principled. As we discuss in Section 5.2, the incorporation of replay buffer experience into the loss function does not change the optimal solution (pi* = expert) (equations 19 and 20). Furthermore, we note that the same does not apply to previous incorporations of the replay buffer in imitation learning (e.g., DAC). The key differentiator is that our regularization is applied to both numerator and denominator of the (implicitly estimated) density ratios, as opposed to replacing the denominator with dRB, as done by DAC.
>
> Regarding overfitting on HalfCheetah and Ant: It is not possible to measure KL exactly in MuJoCo and rewards are used only as a proxy for evaluation of performance. Additionally, the fact that imitation learning minimizes KL between \pi and a small finite sample from \pi_{expert} means that a better imitation learning algorithm (that achieves lower KL) may not recover \pi_{expert} exactly.
>
> Regarding optimization instabilities, in our preliminary experiments, we also found direct application of DualDICE to be difficult to tune, and thus developed a number of remedies to the instability (which may be generally useful). First, we introduced the use of log-average-exp into the variational form of KL (as opposed to the squared loss or f-divergence form suggested by DualDICE). Second, we used gradient penalty regularization, borrowed from previous works on adversarial learning algorithms and dual representations of the KL. We found the combination of these two to provide much more stable performance.
>
> As the reviewer suggested, we are actively working to opensource our implementation.

---

### Official Review · AnonReviewer3 · 2019-10-23
**Official Blind Review #3**

**Rating:** 6

**Review:**

This paper provides a novel off policy objective to solve imitation learning. It resolves the limitation of the famous GAIL algorithm that we need on-policy samples to interact with the environment. The new algorithm is simple but efficient, and can handle off-policy settings. The derivation of equation (12) is nice and intuitive, provide a potential on creating new imitation learning algorithm. Empirical results show that the new algorithm can perform as good as the state-of-the-art baseline, under on-policy setting.

Clarity:
The paper is well written an intuitive. It clearly introduces the previous works and their limitation, and naturally derives the new objective by DualDice trick to resolve the limitation. Section 5 discusses the bias introduce by the exponential of expectation, which in practice does not hurt the performance much. Experimental design is good and informative.

Major concern:
In experiment we only saw the result of on-policy setting using the replay buffer regularizer. However, the first half of the paper focuses on deriving an off-policy objective for imitation learning. A natural question is: how good is the performance if we only use off-policy data? In Figure 3 with enough expert trajectories, how does the off-policy ValueDice perform compared to behavior cloning?

In sum, I think the paper is clearly above the acceptance threshold. But I will leave it 6 point and raise my point if the authors can answer my question above.



**Experience Assessment:**

I have read many papers in this area.

**Review Assessment: Checking Correctness Of Derivations And Theory:**

I carefully checked the derivations and theory.

**Review Assessment: Checking Correctness Of Experiments:**

I carefully checked the experiments.

**Review Assessment: Thoroughness In Paper Reading:**

I read the paper thoroughly.

---

> ### Author Response · Authors · 2019-11-11
> **Re: Official Blind Review #3**
>
> Thanks for the helpful feedback on our paper!
>
> Following the reviewers suggestion, we have included results of offline ValueDICE on the MuJoCo tasks (see Appendix C). Anecdotally, we found that the use of additional online experience is not necessary for the MuJoCo tasks we used for evaluation. We originally focused on the online version because we believe it is more generally useful (and indeed, on the Ring MDP the use of additional experience is crucial to generalize to states not observed in the original expert dataset).

---

> > ### Comment · AnonReviewer3 · 2019-11-13
> > **Thank you for the response**
> >
> > Thank you for the response and the additional experiment.
> >
> > Do you use the same settings as Figure 2? It seems that for Hopper, And and Walker2d the results are not consistent in Figure 2 and 4 for Behavior Cloning method.
> >
> > I am surprised for the implication that "additional online experience is not necessary for the MuJoCo tasks we used for evaluation". I think if you can empirically show that ValueDice can effectively learn expert policy in off-policy setting, this will be extremely useful in many real world task where we cannot collect transition data directly for exploration.

---

> > > ### Author Response · Authors · 2019-11-13
> > > **Re: Thank you for the response**
> > >
> > > For Figure 2, as we mention in the paper, we used the implementation of BC from the original implementation of GAIL. For Figure 4, in order to plot the rewards w.r.t. different number of updates, we use our own implementation. The use of our own implementation also allowed us to borrow the same settings as used for ValueDICE (same network, same optimizer, same regularizations). The results of BC in the Appendix are after appropriate tuning of our implementation over learning rates and regularization weight. In the next revision we will mention this more explicitly.

---

### Public Comment · ~Bronson_T._C.1 · 2020-05-03
**Question about Source Code**

Hi Ilya, thank you for a very interesting paper.

We tried running your source code to replicate the experiments, but encountered a problem. The directory for expert trajectories is specified as follows:

In "run_experiments.sh":

Line 20: expert_dir="./experts/"

However, this directory does not exist. It is stated in the paper that trajectories are taken from GAIL, however neither the data in the official GAIL repo, nor your DAC repo, are of the correct format.

Could you please kindly point us to where we can find the expert trajectories?

Many thanks!

---

> ### Author Response · Authors · 2020-05-08
> **Datasets**
>
> I apologize for the delay.
>
> We updated the repository, you can find the datasets here:
> https://github.com/google-research/google-research/tree/master/value_dice

---

### Decision · Program_Chairs · 2019-12-19

**Decision:**

Accept (Poster)

**Comment:**

This work addresses new insights in the imitation learning setting, and shows how a popular type of approach can be extended in a principled way to the off-policy learning setting. Several requests for clarification were addressed in the rebuttal phase, in particular regarding the empirical evaluation in off-policy settings. The authors improved the empirical validation and overall clarity of the paper. The resulting manuscript provides valuable new insights, in particular in its principled connections, and extension to previous work.